# What arguments and from whom are most influential in shaping public health policy: thematic content analysis of responses to a public consultation on the regulation of television food advertising to children in the UK

Ahmed Razavi,[1] J Adams,[2] Martin White[2,3]

[1]MRC Epidemiology Unit, Cambridge, UK
[2]Centre for Diet and Activity Research, University of Cambridge, Cambridge, UK
[3]Institute of Health and Society, Newcastle University, Newcastle upon Tyne, UK

**Correspondence to**
Dr Ahmed Razavi;
ahmed.razavi@nhs.net

## ABSTRACT

**Objectives** We explore one aspect of the decision making process—public consultation on policy proposals by a national regulatory body—aiming to understand how public health policy development is influenced by different stakeholders.

**Design** We used thematic content analysis to explore responses to a national consultation on the regulation of television advertising of foods high in fat, salt and sugar aimed at children.

**Setting** UK.

**Results** 139 responses from key stakeholder groups were analysed to determine how they influenced the regulator's initial proposals for advertising restrictions. The regulator's priorities were questioned throughout the consultation process by public health stakeholders. The eventual restrictions implemented were less strict in many ways than those originally proposed. These changes appeared to be influenced most by commercial, rather than public health, stakeholders.

**Conclusions** Public health policy making appears to be considered as a balance between commercial and public health interests. Tactics such as the questioning and reframing of scientific evidence may be used. In this example, exploring the development of policy regulating television food advertising to children, commercial considerations appear to have led to a watering down of initial regulatory proposals, with proposed packages not including the measures public health advocates considered to be the most effective. This seems likely to have compromised the ultimate public health effectiveness of the regulations eventually implemented.

## Strengths and limitations of this study

► Established qualitative methodology (thematic content analysis) was used to evaluate all stakeholder responses.
► A de novo analytical framework was created, minimising bias that may have occurred from using a pre-existing framework.
► Stakeholder groups were sorted into eight broad categories, allowing us to compare and contrast responses by category.
► Policy making can be influenced through other non-public means (eg, direct lobbying), making us unable to comment on how other methods of influencing policy making may have affected this consultation's outcome.
► This is one case study of influencing policy and our findings may not be generalisable to other cases.

## BACKGROUND

Foods high in fat, salt and sugar (HFSS) are a contributing factor to increasing rates of non-communicable disease worldwide[1] and the WHO has encouraged member states to take action on non-communicable diseases, including through regulation of the advertising of HFSS foods.[2] However, a 2016 study found that no member states had implemented comprehensive legislation restricting marketing of unhealthy food and beverages to young people,[3] despite multiple systematic reviews demonstrating the importance of food marketing as a driver of childhood obesity.[4–6]

Industry groups often seek to influence public health policy.[7] For example, in 2003, a WHO recommendation suggesting reduction in population sugar intake resulted in the Sugar Association (a sugar industry information group) pressing the US Congress to cut WHO funding.[8] However, influences on dietary public health policy are not limited to the food industry. Health professionals, charities, politicians and members of the public have all attempted to influence policy making. Evidence of the impact of these activities is hard to find in the peer reviewed literature.



| Dec 2003 – The UK government asks Ofcom to consider proposals for strengthening television advertising food aimed at children. | Mar 2006 – Ofcom launches a public consultation with suggestions for restrictions. 1097 responses (114 from 'interested parties') received. | Feb 2007 – Ofcom announces the phasing in of a ban on HFSS advertising during programmes aimed at children to be fully implemented by Jan 2009. |
|---|---|---|

| Nov 2004 – Ofcom's research suggests a modest, direct effect. The Department of Health publish a white paper suggesting a case for restricting promotion of high fat, salt and sugar foods (HFSS). | Nov 2006 – Following the initial consultation an alternative package of advertising restrictions was proposed. 39 responses (25 from 'interested parties') received. |
|---|---|

**Figure 1** A timeline of the Ofcom process on developing new recommendations for limiting television food advertising to children. 'Interested parties' are stakeholder groups who may have been affected by the proposed changes, including advertising agencies, advocacy groups, broadcasters, charities, healthcare associations, politicians, the food industry and the general public. HFSS, high fat, sugar and salt foods.

Systematic reviews[9–11] have demonstrated how the alcohol and tobacco industries focus on lobbying efforts and promote self-regulation as a means to minimise the impact of public health policy on commercial activities. These tactics have also been seen in relation to food where, in one case study, government opinion reflected industry rather than public health opinion.[12] However, at present, we have limited insight into how stakeholders other than those representing industry interests attempt to influence public health policy in general or dietary public health policy in particular. Identifying strategies and arguments used by these interested parties in a public setting may help inform how public health policy is determined and how it might more effectively be developed in the future.

### Policy context

In December 2003, the UK Government asked Ofcom (the UK communications' industry regulator) to consider proposals for strengthening rules on television advertising of food aimed at children (figure 1). Ofcom decided to use the Food Standards Agency's nutrient profiling model to determine which foods were classified as HFSS. Ofcom originally put three proposed 'packages' of regulations to public consultation in March 2006 (packages 1–3 in table 1). Following this, Ofcom produced an alternative package of restrictions (modified package 1 in table 1) in November 2006, on which Ofcom again consulted.

Following the second consultation (November 2006), modified package 1 was recommended by Ofcom and was implemented from January 2009. A comparison of the final regulations implemented to the initial packages proposed suggests that the consultations had substantial impacts on policy decisions. The only independent evaluation of the regulations eventually implemented found no change in the proportion of advertisements seen by children that were for HFSS foods from before to after

implementation, and an increase in exposure of HFSS advertising among adults.[13 14] A '9pm watershed' (ie, no advertising of HFSS foods before 21:00) is now the preferred option of many civil society and public sector organisations to reduce exposure of children to HFSS food advertisings.[15-18]

### Study aims

The consultations on the Ofcom regulations on the restriction of television food advertising to children offers an opportunity to analyse responses from a range

**Table 1** Packages of regulations proposed by Ofcom in the initial consultation (March 2006)

| Option | Details |
|---|---|
| Package 1 | ► No HFSS food advertising during programmes specifically made for children<br>► No HFSS food advertising during programmes of particular appeal to children* aged 4–9 years |
| Package 2 | ► No food or drink advertising during programmes made specifically for children or of particular appeal to children aged up to 9 years |
| Package 3 | ► Volume of food and drink advertising to be limited at times when children are most likely to be watching |
| Modified package 1 | ► As per package 1 except restrictions on HFSS food advertising to be extended to programmes of particular appeal to children aged 4–15 years |

\* 'of particular appeal to children'=when the proportion of people watching who are children is more than 120% of the proportion of children in the UK population.[23]
HFSS, high fat, sugar and salt.

of stakeholder groups to a consultation on an important policy that aims to promote dietary public health through regulation of the food industry. We aimed to identify which arguments, and from which stakeholder groups, appeared to be most influential in shaping the changes in Ofcom's position from the initial consultation to the final recommendations.

## METHODS

We followed the Standards for Reporting Qualitative Research[19] in reporting our findings.

### Patient and public involvement

This study did not involve use of patient identifiable data and only used publicly available responses from stakeholder groups. We did not consult the public on the methods.

### Data sources

We qualitatively analysed all written responses from stakeholder groups to the 2006–2007 Ofcom public consultation on the regulation of television advertising of food and drink to children. The consultation asked for responses to a series of questions regarding the various policy packages outlined by Ofcom. Options such as having a 9pm watershed before which HFSS foods could not be advertised, self-regulation, having a transitional period and exemptions to the regulations were asked about. Responses were freely available on the Ofcom website,[20] and responses to both the first and second consultations were included. Responses from individual members of the public were not included as they tended to be very brief and non-specific. We therefore focused our analysis on key stakeholder organisations representing key constituencies. Where needed, optical character recognition software was used to transcribe the responses. The consultation questions can be seen in the online supplementary table A.

### Data analysis

Conventional thematic content analysis[21] was used to analyse the data, and the Framework method[22] was used to organise and chart the data. This method involves creating coding categories directly from the data and organising coding within a flexible matrix, which can then be adjusted as more codes emerge from the text. As existing literature on the topic of stakeholder influence on public health policy is limited, rather than using preconceived categories with which to code the data, a new framework for analysis was developed, based on no a priori assumptions. After familiarisation with the data, coding was performed line by line for each of the responses from interested parties in NVivo (software developed by QSR International for qualitative research).

Each response was assigned to a category based on the person or organisation from which it originated to stratify responses between the various types of interested parties

**Table 2** Categories into which stakeholder groups were classified

| Category | Definition |
| --- | --- |
| Advertising stakeholders | Advertising companies and representative bodies |
| Broadcast stakeholders | Broadcasting companies and representative bodies |
| Civil society groups | Groups that represent the interests of all or some of the general population. This does not include groups that may have affiliations with industry who would be included in one of the 'stakeholders' groups |
| Food manufacturers | Companies that produce and sell food to retailers |
| Food retailers | A company that sells food to the general population |
| Food industry representative groups | Bodies that represent the interests of groups of food manufacturers and retailers |
| Politicians | Persons professionally involved in politics |
| Public health stakeholders | Groups that focus on promoting the health of the population |

(table 2). These categories were initially determined by assigning labels to each response and then subsequently refined by the reviewers. A list of each group classified by category can be found in the online supplementary tables B1 and B2. The longest and second longest submissions from each category were then coded to develop the initial framework.

Following coding of the first two longest responses in each category by Ahmed Razavi (AR), a set of codes to apply to further responses was agreed between all authors. Codes were also grouped into themes at this stage to provide the most meaningful thematic coding of the data. The remaining responses were all coded using this analytical framework by AR with additional codes being created when needed. Once each of the responses was coded, a 10% sample of the data was independently duplicate coded by one of the other authors (JA or MW) to ensure appropriate categorisation of the various codes and code hierarchy, and to improve internal validity. Using a matrix, the data were charted, resulting in a summary of the data by category from each transcript. Illustrative quotations were highlighted at this point.

The resulting charted data were then interpreted and analysed to determine recurrent themes or topics. These were explored further using quotations to demonstrate the range of opinions in relation to each theme or topic. The positions taken by the interested parties were then compared with Ofcom's starting position and final statement, to identify which positions from which stakeholders appeared to have held the most influence on Ofcom's final position.

## Ethics

Ethical permission was not sought for this study. The consultation responses used have been made freely available on the Ofcom website with the full knowledge of their authors. We, therefore, treat this as publicly available data which does not require ethical permission for analysis. As we did not seek informed consent from the authors of consultation responses, we do not name them here although names were provided on the Ofcom website. Instead, we have used only the categories described in table 2 to identify quotations in our results. This also avoided the study from becoming too focused on specific stakeholders rather than building a general picture of arguments used by different stakeholder groups.

## RESULTS

Of 1136 responses received to both rounds of consultation, 997 were from individual members of the public (and thus excluded from the analysis); 139 were from stakeholder groups and were included in the analysis; 114 were responses to the initial consultation and 25 responses to the second consultation. The vast majority of responses from individuals were one line statements of support for some form of restrictions without directly addressing specific issues concerning implementation. As such it was determined that there was not sufficient detail to determine arguments used, or positions taken.

Therefore, these responses are unlikely to have influenced Ofcom other than to reaffirm that there was public support for some form of restriction.

The stakeholder responses varied in length from a few lines to double digit numbers of pages. Most took the form of an initial broad statement outlining a policy position with supporting evidence, followed by shorter responses directed at addressing the specific questions in the consultation, as outlined by Ofcom (shown in the online supplementary table A).

The organisations in the stakeholder groups outlined in table 2 broadly fell into two separate categories. Civil society groups, politicians and public health stakeholders were encouraging of restrictions in order to reduce the exposure of children to advertising of HFSS foods. Advertising stakeholders, broadcast stakeholders, food manufacturers, food retailers and food industry stakeholders argued that restrictions would minimally impact childhood obesity while having a substantial impact on businesses. Although there were subtleties within each group with regards to what level of restrictions would be ideal, there were not sufficient differences in order to further analyse the differences in responses of the various stakeholder groups beyond these two broad categories.

The key changes from the initial Ofcom position to the final recommendations are summarised in table 3. Arguments relating to each of the principles below, as outlined

| Table 3 | Changes in Ofcom's position during the course of the consultation | | |
|---|---|---|---|
| **Initial options presented by Ofcom** | **Consultation responses and Ofcom's reaction** | **Ofcom's final position** | **Reference in consultation** |
| *Ofcom's packages 1–3 varied on three key principles* | | | |
| (1) Restrictions on advertising of all foods versus just HFSS foods | Following the first consultation it was clear that the majority of responses preferred restricting advertising of only HFSS foods | The eventual package of restrictions enacted was specific to HFSS foods | Ofcom Executive Summary 1.12 |
| (2) Total ban on food advertising versus volume based restrictions | Almost all stakeholders did not consider volume based restrictions as being effective at reducing exposure to advertising and this option was dismissed following the first consultation | There was a total ban enacted on HFSS food advertising in programming 'of particular interest to' children | Ofcom Executive Summary 1.12 |
| (3) Restrictions only on children's channels versus all programmes 'of particular interest' to children, irrespective of channel | Public health and civil society responses highlighted that children may watch adult TV and a ban on all less healthy food advertising before a 9pm watershed may be more effective than focusing specifically on children's programming. Television and advertising industry responses worried that this would disproportionately impact advertising revenues | Ofcom rejected the idea of a pre-9pm ban due to concerns about the effect it would have on broadcasters, programming and advertising revenues | Ofcom Executive Summary 1.12 |
| *Further changes that were made* | | | |
| Restrictions should apply to children aged 4–9 years | Many public health and civil society responses pointed out that children are legally defined as under 16 years | The restrictions applied to children aged 4–15 years | Ofcom Final Statement 4.9 |
| All restrictions should start in April 2007 | Children's channels argued that they should be allowed a transitional period as they would be affected financially | Children's channels were allowed a phased implementation of restrictions, with final implementation by January 2009 | Ofcom Final Statement 5.3/5.4 |

in the recommendations, were captured from the framework and are described in detail.

## To which foods should restrictions apply?

There was non-partisan agreement that having a blanket ban on all television food advertising was counterproductive and had the possibility of inadvertently reducing exposure of children to advertisements for healthier products.

Quotes: Should restrictions apply to all foods?

*"We do not support any options which would restrict advertising of all foods, including foods such as fruit and vegetables, milk and dairy products. These foods can play an important part in children consuming a balanced diet, and we consider that advertising can play a useful role in educating both parents and children in the ways to achieve this."* (Food industry stakeholder)

*"(Public health stakeholder) believes that it is desirable to distinguish between healthy and unhealthy foods. We do not believe it would be useful to restrict the advertising of all foods because this would mean manufacturers and retailers would be unable to promote healthy foods, such as fresh fruit and vegetables."* (Public health stakeholder)

As the underlying aim of the restrictions was to protect health, preventing the advertising of healthy products would be counterproductive. Stakeholder groups agreed that banning advertisements of all foods would be deleterious to efforts to promote healthy eating and promoting a balanced diet.

## Total ban or volume based ban?

The idea of a broad volume based restriction rather than a total ban targeting children's programming was proposed in package 3 and was nearly universally disliked. Broadcasters, advertisers and food industry stakeholders argued that a volume based restriction would have a very large effect on commercial revenues, whereas public health stakeholders and civil society groups cited how little a volume based restriction would actually reduce the exposure of children to HFSS food advertising.

Quotes: Would a volume based restriction be effective?

*"The least acceptable option would be package 3, which would have a devastating effect on our overall revenues — several times greater than Ofcom has estimated— while delivering a smaller reduction in the number of times children see food and drink adverts."* (Broadcast stakeholder)

*"Package 3 not only restricts the option to promote healthy foods to children, but also fails to restrict HFSS adverts during periods of viewing when many children are still watching, that is, up to 9pm."* (Public health stakeholder)

Many responses argued that package 3 would result in very little change in exposure of children to television advertising of HFSS foods but would substantially impact broadcasters and advertisers financially. Arguments concerning commercial impacts were used throughout the responses of industry groups, with emphasis on the

fact that as a broadcast regulator, Ofcom has a duty to minimise impact on revenues for broadcasters.

## Restrictions on children's programming or a pre-9pm watershed ban?

Although not included in any of Ofcom's proposals, one of the consultation questions asked about whether restricting advertising before 9pm would be a suitable measure. In response, civil society groups and public health stakeholders called for restricting all HFSS food advertising before a 9pm 'watershed'. Advertisers, broadcasters and the food industry claimed such restrictions would impinge on adult viewing. All three groups highlighted the trade-off between protecting children and the loss of advertising exposure to adults. Advertisers, broadcasters and food industry groups cited the negative commercial impacts of a pre-9pm watershed ban as outweighing any 'marginal' public health benefits; whereas civil society groups and public health groups saw the public health benefit of a pre-9pm watershed ban as outweighing commercial impacts.

Quotes: Arguments pertaining to the pre-9pm watershed ban on HFSS food advertising.

*"(Food industry stakeholder organisation) welcomes Ofcom's rejection of the pre-9pm watershed, as this would have been tantamount to a complete ban on the advertising of food and soft drink products on television, and would have impacted on adult airtime."* (Food industry stakeholder)

"We believe that the most suitable option is the pre 9pm ban of HFSS advertising, for the following reasons:

► achieves one of the key regulatory objectives, that of significantly reducing the impact of HFSS advertising on younger children

► removes 82% of the recorded HFSS advertising impacts on all children (aged 4–15 years)

► contributes substantially to enhancing protection for older children by reducing their exposure to HFSS advertising

► offers the greatest social and health benefits of all options, in the ranges of £50 million to £200 million per year or £250 million to £990 million per year (depending on the value of life measure)". (Civil society group)

*"The avoidance of intrusive regulation of advertising during adult airtime is only justifiable once full account has been taken to address the overriding priority to protect children's health. At times when adults and children are watching, the need to protect children must take priority."* (Public health stakeholder)

In their final statement following the consultation,[23] Ofcom explained why they had rejected banning HFSS food advertising before a 9pm watershed due to the effect this was expected to have on adult viewing times and commercial revenues. Industry groups appeared to be successful in arguing that adult viewing should be

unaffected despite the possibility that both children and adults may be watching television together. The need to protect the right of adults to see whatever they wish was a common argument against restricting advertising on television channels that were not explicitly targeted at children. The individual freedom of an adult therefore appeared to be given precedence over exposing children to HFSS food advertising.

Ofcom's research[23] showed that 48% of parents supported restricting HFSS food advertising before 9pm, which was often cited by industry responses as evidence of a lack of public support. Some responses highlighted the fact that the complete figures were 48% in support of a pre-9pm watershed ban, 24% against the ban, with the remainder undecided. An apparently valid complaint made by public health groups regarding this issue was that Ofcom did not ever consult on a pre-9pm watershed ban despite its own research showing this would reduce the exposure of children to HFSS advertising by 82%.

We are also able to see here the use of evidence based arguments by the civil society group in making their case. Some civil society groups and public health stakeholders would cite evidence to support their argument. The quotes above illustrate an example of how a civil society group used data and evidence to support their arguments by, for instance, suggesting that banning advertising prior to 9pm could reduce advertising exposure of children by 82%. This figure was taken from Ofcom's own analysis of the effects of the various policy options, which can now be found included in Ofcom's final report on the consultation.[23] Food industry representative groups on the other hand tended to cite a lack of evidence or only used evidence that appeared to support their arguments.

Quotes: Arguments regarding available evidence and its interpretation.

"*As Ofcom has found from its own research, television advertising has only a 'modest direct effect' on children's food preferences, consumption and behaviour, and that other factors—including taste, price familiarity, peer pressure and convenience—all have a higher effect. Hastings, in his report for the Food Standards Agency, found that advertising had only a 2% direct effect on children's choice.*" (Food company)

"*Ofcom quotes an estimate that advertising/television accounts for some 2% of variation in food choice/obesity. This is not a small figure considering that calculations by the Institute of Medicine show that this would mean an estimated additional 1.5 million young people in the US falling into the obese category.*" (Public health interests)

"*The evidence that television has anything but an extremely small impact on the HFSS element of the diet of children is unconvincing and accordingly it is difficult to support proposals that appear disproportionate.*" (Broadcast interests)

## To what ages of children should the restrictions apply?

Ofcom initially planned to restrict advertisements targeted at children aged 4–9 years, although this was subsequently expanded to cover children ages 4–15 years in the final regulations. Children under 4 years were thought to have little influence over what foods and drinks were given to them and therefore not considered as part of the restrictions. Throughout the consultation, food industry representative groups and food manufacturers argued that restricting advertisements to children aged 4–9 years was appropriate, whereas as public health stakeholders argued that this should be expanded to cover children aged 4–15 years.

Quotes: Arguments pertaining to the age of children to which restrictions should apply.

"*It is neither logical nor is there any explanation as to why Ofcom should propose to limit the focus of regulation to children aged under 10 years. The government asked Ofcom to consider proposals for strengthening its rules on television advertising of food to children. It did not ask Ofcom to limit its focus to any particular age group. Ofcom should logically apply restrictions according to its own definition of children (aged 15 years (or under)).*" (Public health stakeholder)

"*Children develop and refine their ability to interpret advertising messages as they get older. Existing studies suggest that by 10 years old (indeed, most studies suggest an even earlier age) they are considered to have sufficient cognitive development to understand the implications of television advertising.*" (Food manufacturer)

"*We are alarmed by the decision to extend volume and scheduling restrictions of food and drink advertising to children under 16 years. The intention of Ofcom and the government has always been to protect younger children and industry responded on this basis. Ofcom has previously stated that it wished to find a proportionate solution and we question the evidence base on which this decision was made. A review of Ofcom's own literature would seem to contradict the question put to consultation and support the conclusion that young people are capable of differentiating between programming and advertising.*" (Food industry representative group)

The logic of defining children as aged 4–9 years was questioned by many stakeholders as, according to Ofcom and in the UK, children are legally defined as those under the age of 16 years. A number of food manufacturers stated that they already did not advertise their products to children under 8–12 years. They argued that during adolescence children become 'media literate' and are able to understand advertising and should therefore not be a target of the restrictions.

Industry arguments appeared to suggest that media 'illiterate' children need protecting from HFSS food advertising whereas public health groups suggested all children needed protecting regardless of how 'media literate' they are. Public health groups argued that adolescents are still susceptible to advertising, have more

purchasing power and greater pester power than younger children, and may not appreciate the health implications of a poor diet. Ofcom concluded that expanding restrictions to include children aged 4–15 years was appropriate, suggesting the arguments of public health groups held more weight over this issue.

### When should the restrictions start?

The need for a transitional period was also hotly debated. Public health stakeholders and civil society groups suggested that as companies were already aware that restrictions were due to be enforced any transitional period should be minimal. Industry groups argued that a transition period was necessary to allow adjustments to be made.

Quotes: Arguments pertaining to the need for a transitional period.

> *"We do not believe [a] transitional period is appropriate. The arguments for 'phasing in' restrictions appear to be of a commercial nature and not supportive of the policy's public health objectives."* (Public health stakeholder)

> *"We would ask for a transitional period of at least 3 years. This would allow production companies to adjust, and the growing number of public companies to issue profit warnings where necessary."* (Broadcast stakeholder)

Instead of starting restrictions soon after announcement of the final policy statement (February 2007), a phased transition over 1–2 years was implemented (varying for different channel types), suggesting industry arguments held more weight on this point. Despite the stated objective of minimising the exposure of children to HFSS food advertising, it appears that Ofcom was more concerned about the potential commercial impact of advertising restrictions and delayed enforcement of the restrictions as a result.

## DISCUSSION

### Summary of principal findings

This study presented a unique opportunity for a detailed analysis of responses to a public consultation on a public health policy in the UK. Such data are often not in the public domain and these data therefore offered a rare opportunity for scientific scrutiny. For example, verbatim responses to the 2016 consultation on the UK Soft Drinks Industry Levy have not been released. Our paper highlights how, despite the relative transparency of the 2006–2007 consultation, the final policy appeared to be substantially influenced by stakeholders. Commercial and public health interests aligned with regards to whether restrictions should apply to all foods or just HFSS foods as neither wished to ban advertising of healthy foods. Likewise, common ground was found when considering a volume based ban, with it having large commercial impact but little public health impact as per Ofcom's own findings.[23]

As far as we are aware, this is the first analysis to examine how a range of stakeholder groups influenced the development of a public health policy aiming to regulate food industry advertising. Ofcom's decision to implement modified package 1 contained concessions to commercial as well as civil society and public health stakeholders. However, ultimately, industry arguments appeared to hold more sway, with the main concession to public health groups being expanding restrictions from children aged 4–9 years to those aged 4–15 years. Ofcom appeared to believe that the commercial impact of the regulation of advertising should carry greatest weight, even when the aim of the regulation was to protect children's health. As such, Ofcom did not formally consider a pre-9pm ban as part of any of its packages, as had been proposed by public health and civil society stakeholders, although one of the consultation questions did refer to a pre-9pm ban. Instead, Ofcom approved a 2 year transition period and emphasised the need for 'proportionate action'. Some responses to the consultation from public health advocates argued that Ofcom, being a broadcast regulator rather than a public health stakeholder, felt an obligation to protect industry interests. The case for restricting advertising was made in a Department of Health 'white paper'[24] (NHS strategy documents are known as 'white papers'). However, Ofcom was tasked with determining how to implement these restrictions. Under the Communications Act 2003, Ofcom retains direct responsibility for advertising scheduling policy. This then begs the question of whether a governmental body with a duty to protect broadcasting interests should be leading on public health legislation.

This conflict between Ofcom's duties to the public and to broadcasters may have resulted in eventual restrictions that did not appear to alter the level of exposure of children to HFSS food advertising.[13 14] Ofcom appeared to balance arguments related to commercial and public interests, in terms of jobs and the wider economy, with those relating to public health. Being proportionate in their restrictions was frequently cited by Ofcom in their decision making. Ofcom did not, however, appear to consider the cost to the economy of poor health that could stem from a lack of appropriate restrictions. Although this was cited by some public health groups (see quotes pertaining to a pre-9pm ban) this does not appear to have been considered by Ofcom in their final report, with no mention of wider societal costs. Ofcom also appeared to give greater priority to allowing advertisers access to adults than to restricting exposure to HFSS food advertising among children, who may be viewing the same programming. Industry representative groups tended to highlight commercial arguments while citing evidence that appeared to downplay the role of television advertising in childhood obesity. Public health groups emphasised that the health of children should outweigh any financial concerns and pointed out that even small changes to advertising at an individual level would affect large numbers of children and so accrue to large population level benefits.

## Strengths and limitations

Using established qualitative methods allowed us to identify key themes in the consultation responses according to stakeholder interests. The creation of a de novo framework minimised bias that might have been imposed by using a pre-existing framework. Instead, we allowed categories to emerge from the data. The classification of the responses also enabled us to see what positions were taken by the various stakeholders and which type of responses carried the most influence. Measures were taken to maximise the reliability of our coding, such as duplicate coding a sample of consultation responses. The use of publicly available data was resource efficient. Additionally, the use of all the available data ensured that no perspectives were omitted, adding to internally validity. The omission of responses from individual members of the public was because most public responses lacked detail and were no more than a sentence long. Commercial influences on public health policy are unlikely to have changed over the past decade with no changes in lobbying rules or policy making procedures, making it highly likely that our findings from the 2007 consultation are applicable today.

There may be alternative methods by which the public influences policy making, such as by writing to their member of parliament. This is a study of only one case of public health policy making and our specific findings may not be generalisable to other aspects of dietary public health policy specifically or public health policy more generally. In this consultation, all members of a stakeholder category were treated as one, although there was some inter-category variation on position. A cross question analysis could have been performed analysing responses by each question posed, although many of the responses were free text and did not address each question directly. In this study, we have only addressed what arguments and from whom are most influential in shaping public health policy, not specifically the various methods by which different stakeholders influence policy. There are also other ways by which interested parties could influence Ofcom, which we were unable to examine in this study. For example, Ofcom gave the option of providing confidential responses which were not available for us to incorporate into our dataset. Other informal lobbying may have occurred. Whether such channels of influence were used or whether similar arguments will have been used privately as were used publicly is unclear. Further work could explore other means of influence in due course.

## Relationship to existing knowledge

Some literature exists on the methods by which public health advocates influence policy. In 2006, the New Zealand government held an 'inquiry into obesity' in order to determine what could be done to limit increasing obesity rates.[12] Jenkin *et al* found that in three out of four domains examined, the governmental position aligned with that of industry groups, with the exception being nutritional policy in schools. In the other three domains, national obesity strategy, food industry policy,

and advertising and marketing policy, the analysis determined that the governmental position allied with industry groups. Much like our study, public health groups were shown to have a limited impact on the eventual policies, with industry arguments proving more influential. An explanation suggested for this was the significance of the food industry to New Zealand's economy, highlighting how considerations outside of public health may importantly shape public health policy. It may be the case that similar factors shaped the eventual restrictions in our case study, despite the appearance of Ofcom wanting to develop 'proportionate restrictions', balancing commercial and public health interests. The question of what is proportionate appears to be determined by ideology and how much one feels government's role is to protect health even if it impacts on industry. If this is the case, we must question whether commercial companies can ever be truly motivated to improve health at the possible detriment to their short term profits. A thematic analysis of alcohol industry documents in Australia[25] concluded that the industry attempted to create an impression of social responsibility while promoting interventions that did not affect their profits and campaigning against effective interventions that might affect profits. The de facto exemption of commercial stakeholders from bearing the negative external costs of their profitable endeavours (eg, environmental, social or health impacts) has been widely questioned.[26]

## Interpretation and implications of the study

Much of the research undertaken to date on stakeholder influences on public health policy has focused on industry behaviours and practices, whereas in this study we have treated both pro-industry and pro-public health groups equally in our analysis. Industry groups were apparently successfully able to argue that extensive restrictions would impact on their commercial revenues, suggesting that their economic arguments importantly influenced the thinking of policy makers. However, the future (external) costs of treating the potential health implications of HFSS food consumption did not appear to influence policy making. This may be because any potential cost savings are long term and would apply to the health sector, for which Ofcom has no governmental responsibility, whereas the short term costs would apply to the broadcast sector for which Ofcom is the regulatory body.

Public health advocacy is an activity in which many public health professionals are keen to become more effective to better ensure that evidence is translated into policy.[27 28] This study highlights that responding to public health policy consultations alone may not result in policy making favourable to public health and other avenues of influence may also need to be explored. Conversely, the change in the definition of children from 4–9 years to 4–15 years demonstrates that there is scope for public health advocates to shape policy should an issue be sufficiently clear and difficult to oppose. A more Machiavellian

interpretation would be that to define children as aged 4–9 years at the outset may have been a cynical ploy aimed at ensuring that there was at least some ground to concede to public health stakeholders and distract from the more contentious issues. This is supported by the fact that the definition of children as aged 4–9 years was inherently questionable, given Ofcom's own definition of children as under 16 years, in line with the legal and medical definitions used in the UK. A few companies pointed to their media literacy campaigns as evidence that adolescents can understand advertising as an argument against redefining the scope of these restrictions to children aged 4–15 years. Evidence shows that advertisers simply use different ways to target adolescents,[29] rendering media literacy moot,[30] and suggesting that restrictions are still needed to protect adolescents.

The issue of TV advertising of less healthy foods remains highly politically sensitive and at the top of the public health strategy agenda for obesity.[18] Many UK public health organisations have recently campaigned to ban television advertising of less healthy foods before 9pm (the so-called 9pm watershed).[16 17 31–34] Our analysis of the 2006–2007 consultation offers specific insights that could be influential in this ongoing national debate, in the same way as such analyses of historical documents have influenced tobacco control efforts in recent years.[10 35] The Ofcom regulation of television advertising of less healthy foods to children is one of few national public health policies of this sort to have been independently evaluated.[14 36] The independent evaluation found that the introduction of the regulations were not associated with a decrease in children's exposure to less healthy food advertising.[36] Our analysis sheds further light on why and how a regulatory policy that appears to have been ineffective in reducing children's exposure to less healthy food advertising came about. Publishing responses to public consultations in full is a key component of transparent policy making. The UK treasury's reluctance to make available responses to the Soft Drinks Industry Levy consultation is contrary to this principle.

### Further questions and future research
How policy making is influenced through means other than public consultations should be further studied. Other means of applying political pressure such as political lobbying and having indirect relationships with positions of power are much more opaque and difficult to monitor. Parliamentary records of lobbying activity, copies of internal or leaked documents, and registers of interests of members of parliament may all be potential sources of data to explore these issues further. Interviews with former or current employees of policy forming bodies such as Ofcom, as well as other stakeholder groups, could also be fruitful. Claims made during this consultation, such as industry claims of needing to issue profit warnings as a consequence of lost revenue from these restrictions, could be analysed. Thematic analysis of further documents, such as the responses analysed in this study,

could provide valuable insight into whether a similar combination of commercial arguments and questioning scientific data is used across different public health policy consultations.

### CONCLUSION
This analysis increases our understanding of how influential some stakeholders are in policy making and provides a framework from which further understanding of the influences on public health policy can be determined. From this case study, we can see that commercial influences on dietary public health policy making appear to be somewhat greater than the influence of public health stakeholders in the initial framing of the consultation and this imbalance may have contributed to the ultimately compromised legislation. In this case, the potential for commercial impacts of legislation promoting public health appeared to outweigh the anticipated population health benefits in policy decision making.

**Contributors** Responses were coded by AR with a subsample independently duplicate coded by JA or MW. AR, JA and MW contributed to the manuscript in terms of both writing and editing.

**Funding** This work was undertaken by the Centre for Diet and Activity Research (CEDAR), a UKCRC Public Health Research Centre of Excellence. Funding from Cancer Research UK, the British Heart Foundation, the Economic and Social Research Council, the Medical Research Council, the National Institute for Health Research, and the Wellcome Trust, under the auspices of the UK Clinical Research Collaboration, is gratefully acknowledged.

**Competing interests** None declared.

**Patient consent for publication** Not required.

**Provenance and peer review** Not commissioned; externally peer reviewed.

**Data sharing statement** Responses to the consultation were freely available on the Ofcom website.

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
