## [Reviewer comments · BMJ Open]

ARTICLE DETAILS

TITLE (PROVISIONAL)	What arguments and from whom are most influential in shaping public health policy: Thematic content analysis of responses to a public consultation on the regulation of television food advertising to children in the UK
AUTHORS	Razavi, Ahmed; Adams, J; White, Martin

VERSION 1 - REVIEW

REVIEWER	Deborah Bowen University of Washington USA
REVIEW RETURNED	04-Jan-2019

GENERAL COMMENTS	This manuscript needs a bit of reorganization for optimal reading. Here is a series of things to attend to When the respondent is a group, how do you know? just simply define it in the text Figure 1 was not in the version of the manuscript I reviewed There is too much detail in the introduction. Simply state the review process and not really the history readers get lost in the details. The introduction does not exactly lead up to the research questions, which should be specified in the end of the intro. What were the research questions of interest and how does this manuscript answer them. Right now it is a description of events. The authors used Coreq reporting guidelines but not all the materials were in the manuscript and I did not see the reporting checklist. what were the different stakeholders identified and how were these categories come by? a table in the appendices with definitions might be helpful The entire analysis is done by period and by company responses. But the title and abstract imply that the entire analysis should be done by stakeholder. So, when you specify research questions make sure they align with the way in which the results are presented. Don't put new data in the discussion and the first entire page runs into this. Please analyze the data and report it in the results and use the discussion to discuss those data.
---

REVIEWER	Anne Marie Thow University of Sydney
REVIEW RETURNED	11-Jan-2019

GENERAL COMMENTS	This paper presents the findings of a qualitative content analysis of consultation submissions regarding marketing restrictions in the UK. The study uses a robust method for content analysis. However, I have serious concerns regarding the interpretation and presentation of findings, which are detailed below. First, the structure of the findings and discussion seems to limit the analytical value of the study and its contribution to the literature. Second, the conclusion of the study that industry actors have had major and disproportionate influence on the policy outcome do not seem to me substantiated by the findings that are presented. Method & Results 1. This is a relatively small study – with a very limited sample and a narrow focus on a single intervention. What the study does, it does well. The rationale for the study and the approach taken is clearly explained. The authors are very up front about the fact that public consultation is only one way in which stakeholders influence policy. However, the combination of this extremely narrow focus (both in intervention and in the policy process), together with – as I note in detail below – the lack of nuance in the analysis and interpretation of findings, means that I am not convinced that it makes a significant contribution to the literature. 2. Minor points:  • Would be useful to briefly summarise the gist of the consultation questions in the method • Line 181 – should the first use of the term ‘stakeholder groups’ in this sentence be instead ‘specific stakeholders’? • Lack of specificity in the number of stakeholders making certain arguments. The use of “few” or “some” seems unnecessary, given the approach to data analysis employed (the authors surely have specific numbers). • Table on page 10 is not specific enough to be useful – the paper argues that there are differences by stakeholder groups and this should be clear in the summary table: e.g. “Some responses highlighted that children may watch adult TV... Other responses worried that this would disproportionately impact advertising revenues” (emphasis added) Findings 3. Structuring the findings to address only the consultation questions separately I think minimises the analytical value of the study. This doesn’t seem to stray far from what an interested observer could see from the submissions and Ofcom’s response. I think the current examination of the policy change in relation to what stakeholders were requesting is essential, but that the study would add more value with additional inclusion of a cross-question analysis, that examines the core study question of how stakeholders influence policy. For example: Line 304-305: “We are also able to see here the use of evidence-based arguments by the civil society group in making their case... [etc]” – this seems a really pivotal point in terms of ‘how’ stakeholders use evidence to seek to influence policy, but
--

has no specificity or detail (i.e. nuance) and is not substantiated by any quotes. What evidence? Relevant to the UK? Was it the same evidence noted in line 307, that industry was seeking to downplay? Or different evidence? Also, was there no difference among “the civil society group” (presumably non-homogenous) in their use of evidence? Could you give some examples?

4. Lines 290-291: “Ofcom rejected banning HFSS food advertising before a 9pm watershed due to the effect this was expected to have on adult viewing times and commercial revenues, suggesting that industry arguments were more persuasive on this topic”
Unless I completely misunderstand the sequence of events as stated in lines 259-260, this rejection occurred prior to the consultation, and thus cannot be explained by the data presented here? As a result, I don’t think that statements like the following should appear in the results: line 290 “Ofcom rejected banning HFSS food advertising before a 9pm watershed due to the effect this was expected to have on adult viewing times and commercial revenues”. But if the authors want to make this assertion, it needs to be referenced. The two paragraphs following this statement don’t seem to me to be presentation of the findings of this study (relating to the evidence base for the 9pm watershed). They don’t arise from the data analysed in the study.

5. Line 310: “Ofcom initially planned to restrict advertisements targeted at children aged 4-9 years”
Please follow this statement with a note on the final policy decision made, as you do in other sections. It’s hard to interpret the stakeholder commentary without reference to the final policy – i.e. move info from line 350: “Ofcom concluded that expanding restrictions to include children aged 4-15 was appropriate.”

6. The authors seem to draw conclusions about industry influence but not public health influence, which seems to me to indicate a bias regarding industry influence. For example, in the section on the change in age group to be much broader than originally proposed (as advocated by public health actors) there is no comment made about public health influence. In contrast, in the section on the phased period for implementation (1-2 years, which I suspect was less than what industry was asking for), the authors state, line 369: “Instead of starting restrictions soon after announcement of the final policy statement (February 2007), a phased transition over 1-2 years was implemented (varying for different channel types), suggesting industry arguments held more weight on this point.”

Discussion

7. Line 382: “policy appeared to be substantially influenced, most importantly by commercial stakeholders”
I’m not convinced the analysis substantiates this point

- The first two questions had agreement across different stakeholder groups. This seems of significant interest to me – and is completely absent from the discussion
- The 9pm watershed was not part of Ofcom’s proposal – but this is one of the key points raised by authors in justifying the impact of industry over public health. It wasn’t on the table, so how can it be evidence of Ofcom changing policy to reflect stakeholder influence in the consultation?

	 • There is no mention of the public health 'win' with the definition of children being expanded to 15 years 8. Similarly, the authors statement in line 392-4 regarding Ofcom's preference for industry does not seem to reflect the results as presented here. "Ofcom appeared to believe that the commercial impact of the regulation of advertising should carry greatest weight, even when the aim of the regulation was to protect children's health. As such, Ofcom rejected a pre-9pm ban, as proposed by public health and civil society stakeholders". My reading of the results is that on the 9pm ban, a decision was made prior to this consultation, so it sits outside of the scope of this analysis. The industry desire for a transitional implementation period (note this is in line with, for example, the World Trade Organization's TBT Committee's requirements for good regulatory practice) was taken into account, but so was the public health community's request for a larger age group. The other two points seem to have had consensus across groups. 9. line 411: "Ofcom did not, however, appear to consider the cost to the economy of poor health that could stem from a lack of appropriate restrictions."Please clarify whether this arose in the consultation – stated like this it's hard to see the rationale for this statement – see also my comment above on providing a more nuanced account of the use of evidence: 10. I don't understand the relevance of the comment on self regulation from line 416 onwards, as this is not self regulation. Please provide a more nuanced discussion of this regulation in light of other international approaches instead. 11. The section titled "Relationship to existing knowledge" from line 452 onwards should be integrated into the discussion, not presented separately, as should the section titled "Interpretation and implications of the study", which duplicates the discussion. I would suggest that the authors use subheadings more strategically throughout the Discussion section, to highlight the key messages and implications, and to reflect on opportunities to potentially strengthen the consultation approach so that public health considerations can be front and centre.
--	--

VERSION 1 – AUTHOR RESPONSE

Reviewer 1:

When the respondent is a group, how do you know? just simply define it in the text

- Page 7, line 155 and 156 clarify how each response was categorised.

There is too much detail in the introduction. Simply state the review process and not really the history readers get lost in the details.

The introduction does not exactly lead up to the research questions, which should be specified in the end of the intro. What were the research questions of interest and how does this manuscript answer them. Right now it is a description of events.

- We have reduced the length of the introduction as suggested. However, we believe that the policy context remains highly relevant for placing this article in context. We have highlighted the study aims

with an additional subheading to clarify that the introduction very clearly leads up to the aims of the paper. In this case we prefer to state aims rather than research questions. As stated, our aims were to identify which arguments, and from which stakeholder groups, appeared to be most influential in shaping the changes in Ofcom's position from the initial consultation to the final recommendations.

The authors used Coreq reporting guidelines but not all the materials were in the manuscript and I did not see the reporting checklist.

- We used the SRQR checklist and this was included in the submission. We will ensure that this is done for the re-submission as well.

What were the different stakeholders identified and how were these categories come by? A table in the appendices with definitions might be helpful

- Table 2 outlines the definitions for each of the stakeholder categories. We have added a sentence to explain how the categories arose (line 157 page 7)

The entire analysis is done by period and by company responses. But the title and abstract imply that the entire analysis should be done by stakeholder. So, when you specify research questions make sure they align with the way in which the results are presented.

- We use the term 'stakeholder group' in the title and abstract (page 2, line 40) to represent the various companies, civil society groups, and representative bodies. We consider the results section to reflect this as we have presented quotes and results by stakeholder group.

Don't put new data in the discussion and the first entire page runs into this. Please analyze the data and report it in the results and use the discussion to discuss those data.

- In the first page of the discussion we summarise the main findings presented in the results section and begin to interpret these and put them into a policy context. This conforms with BMJ Guidelines. We do not believe we have introduced any new data.

Reviewer: 2

Minor points:

- Would be useful to briefly summarise the gist of the consultation questions in the method

- Added, page 6-7 lines 138-140

- Line 181 – should the first use of the term 'stakeholder groups' in this sentence be instead 'specific stakeholders'?

- Changed as above.

- Lack of specificity in the number of stakeholders making certain arguments. The use of "few" or "some" seems unnecessary, given the approach to data analysis employed (the authors surely have specific numbers).

- Table on page 10 is not specific enough to be useful – the paper argues that there are differences by stakeholder groups and this should be clear in the summary table: e.g. "Some responses highlighted that children may watch adult TV... Other responses worried that this would disproportionately impact advertising revenues" (emphasis added)

- This was a qualitative analysis and, as such (and as stated in the aims), we were interested in the range of arguments made by different stakeholder groups and which of these appeared to be most influential in shaping eventual policy. We were not interested in the frequency with which different arguments were made and it was not our intention to conduct a quantitative analysis of the relationship between, for example, frequency of argument and likelihood of that argument shaping policy. Thus, we have chosen not to include specific numbers in the text, as we think this would detract from the qualitative nature of the enquiry and not add useful information.

Findings

3. Structuring the findings to address only the consultation questions separately I think minimises the analytical value of the study. This doesn't seem to stray far from what an interested observer could see from the submissions and Ofcom's response. I think the current examination of the policy change in relation to what stakeholders were requesting is essential, but that the study would add more value with additional inclusion of a cross-question analysis, that examines the core study question of how stakeholders influence policy.

For example: Line 304-305: "We are also able to see here the use of evidence-based arguments by the civil society group in making their case... [etc]" – this seems a really pivotal point in terms of 'how' stakeholders use evidence to seek to influence policy, but has no specificity or detail (i.e. nuance) and is not substantiated by any quotes. What evidence? Relevant to the UK? Was it the same evidence noted in line 307, that industry was seeking to downplay? Or different evidence? Also, was there no difference among "the civil society group" (presumably non-homogenous) in their use of evidence? Could you give some examples?

- Lines 315-319 added to show an example of how evidence was used by a civil society group. The evidence used in this case was Ofcom's own findings, which are referenced. Language of the text has also been adjusted to demonstrate that some responses cited evidence whereas others did not.

4. Lines 290-291: "Ofcom rejected banning HFSS food advertising before a 9pm watershed due to the effect this was expected to have on adult viewing times and commercial revenues, suggesting that industry arguments were more persuasive on this topic"

Unless I completely misunderstand the sequence of events as stated in lines 259-260, this rejection occurred prior to the consultation, and thus cannot be explained by the data presented here? As a result, I don't think that statements like the following should appear in the results: line 290 "Ofcom rejected banning HFSS food advertising before a 9pm watershed due to the effect this was expected to have on adult viewing times and commercial revenues". But if the authors want to make this assertion, it needs to be referenced. The two paragraphs following this statement don't seem to me to be presentation of the findings of this study (relating to the evidence base for the 9pm watershed). They don't arise from the data analysed in the study.

- This has now been clarified (line 270 and line 300) to highlight that, although Ofcom did not make restricting advertising before 9pm part of any of their packages, one of the consultation questions stakeholders could respond to was on the 9pm restriction. A reference has been added to show where Ofcom's rationale for this decision arose from.

5. Line 310: "Ofcom initially planned to restrict advertisements targeted at children aged 4-9 years"

Please follow this statement with a note on the final policy decision made, as you do in other sections. It's hard to interpret the stakeholder commentary without reference to the final policy – i.e. move info from line 350: “Ofcom concluded that expanding restrictions to include children aged 4-15 was appropriate.”

- Clarification added line 315

6. The authors seem to draw conclusions about industry influence but not public health influence, which seems to me to indicate a bias regarding industry influence. For example, in the section on the change in age group to be much broader than originally proposed (as advocated by public health actors) there is no comment made about public health influence. In contrast, in the section on the phased period for implementation (1-2 years, which I suspect was less than what industry was asking for), the authors state, line 369: “Instead of starting restrictions soon after announcement of the final policy statement (February 2007), a phased transition over 1-2 years was implemented (varying for different channel types), suggesting industry arguments held more weight on this point.”

- Line 370 added to make explicit that with regards to the definition of children, public health arguments held more weight.

Discussion

7. Line 382: “policy appeared to be substantially influenced, most importantly by commercial stakeholders”

I'm not convinced the analysis substantiates this point

- The first two questions had agreement across different stakeholder groups. This seems of significant interest to me – and is completely absent from the discussion

- The 9pm watershed was not part of Ofcom's proposal – but this is one of the key points raised by authors in justifying the impact of industry over public health. It wasn't on the table, so how can it be evidence of Ofcom changing policy to reflect stakeholder influence in the consultation?

- There is no mention of the public health 'win' with the definition of children being expanded to 15 years

- Lines 401-406 added to highlight the agreement on the first two questions

- Line 417 added to show that the fact that a 9pm ban was not even considered is evidence in itself that industry interests were favoured

- Line 413 already mentions the public health 'win'

8. Similarly, the authors statement in line 392-4 regarding Ofcom's preference for industry does not seem to reflect the results as presented here. “Ofcom appeared to believe that the commercial impact of the regulation of advertising should carry greatest weight, even when the aim of the regulation was to protect children's health. As such, Ofcom rejected a pre-9pm ban, as proposed by public health and civil society stakeholders”. My reading of the results is that on the 9pm ban, a decision was made prior to this consultation, so it sits outside of the scope of this analysis. The

industry desire for a transitional implementation period (note this is in line with, for example, the World Trade Organization’s TBT Committee’s requirements for good regulatory practice) was taken into account, but so was the public health community’s request for a larger age group. The other two points seem to have had consensus across groups.

- Ofcom did ask about the 9pm watershed as part of the consultation, a point which has been made clearer through additions in line 270, line 300 and lines 437-438. This would therefore be considered part of the scope of this analysis as responses to this question were analysed.

9. line 411: “Ofcom did not, however, appear to consider the cost to the economy of poor health that could stem from a lack of appropriate restrictions.” Please clarify whether this arose in the consultation – stated like this it’s hard to see the rationale for this statement – see also my comment above on providing a more nuanced account of the use of evidence:

- Line 434-437 have been added to give an example of the cost to the economy data cited in a consultation response and the lack of recognition of the costs to wider society in Ofcom’s final report.

10. I don’t understand the relevance of the comment on self regulation from line 416 onwards, as this is not self regulation. Please provide a more nuanced discussion of this regulation in light of other international approaches instead.

- Comment on self-regulation removed.

11. The section titled “Relationship to existing knowledge” from line 452 onwards should be integrated into the discussion, not presented separately, as should the section titled “Interpretation and implications of the study”, which duplicates the discussion. I would suggest that the authors use subheadings more strategically throughout the Discussion section, to highlight the key messages and implications, and to reflect on opportunities to potentially strengthen the consultation approach so that public health considerations can be front and centre.

- The sub-headings are as suggested in BMJ Open instructions to authors, which we have followed.

VERSION 2 – REVIEW

REVIEWER	dEBORAH Bowen University of Washington, USA
REVIEW RETURNED	27-Mar-2019

GENERAL COMMENTS	Authors have done a great job responding to the critiques of the first round. Could they consider one more thing? The title is rather generic and could be made more interesting and specific to the issue and methods
--

REVIEWER	Anne Marie Thow University of Sydney
REVIEW RETURNED	11-Apr-2019

GENERAL COMMENTS	The paper has been strengthened in several regards, however, I still have some concerns. In particular, I do not see that the authors have engaged with the substance of the concerns that I raised in my previous review. Although the authors have acknowledged the major concern I raised regarding the
--

conclusion, and amended the body of the article to make the public health influence clearer, the overall conclusion still states "Public health policy making may prioritise commercial over public health interests" rather than reflecting the findings of this study; the data as far as I understand it shows the influence of both public health and industry, and minimal 'watering down' of the policy. As per Table 1, it appears the main change from Package 1 (the strongest regulation proposed) to Amended Package 1 (which was adopted), was in fact to strengthen it in line with public health.

Other unaddressed concerns are detailed below.

1. My query pasted below did not seem to be clear - what I was referring to was the need to note the differences between stakeholder groups in the table, not to add numbers (ie. "some respondents" doesn't give an indication of stakeholder group differences). This is in line with the articulated aim of the study (p6): "We aimed to identify which arguments, and from which stakeholder groups, appeared to be most influential in shaping the changes of Ofcoms position from the initial consultation to the final recommendations." The data as presented in Table 3 does not give any indication of which stakeholder groups made which arguments.

**** previous comment and response ****

- Table on page 10 is not specific enough to be useful – the paper argues that there are differences by stakeholder groups and this should be clear in the summary table: e.g. "Some responses highlighted that children may watch adult TV... Other responses worried that this would disproportionately impact advertising revenues" (emphasis added)

- This was a qualitative analysis and, as such (and as stated in the aims), we were interested in the range of arguments made by different stakeholder groups and which of these appeared to be most influential in shaping eventual policy. We were not interested in the frequency with which different arguments were made and it was not our intention to conduct a quantitative analysis of the relationship between, for example, frequency of argument and likelihood of that argument shaping policy. Thus, we have chosen not to include specific numbers in the text, as we think this would detract from the qualitative nature of the enquiry and not add useful information.

**** end previous comment and response ****

2. There was no response given to the substantive suggestions in the following queries - in particular, the need for additional discussion that relates to international experience and a clearer logic throughout the discussion.

**** previous comment and response ****

10. I don't understand the relevance of the comment on self regulation from line 416 onwards, as this is not self regulation. Please provide a more nuanced discussion of this regulation in light of other international approaches instead.

- Comment on self-regulation removed.

	11. The section titled “Relationship to existing knowledge” from line 452 onwards should be integrated into the discussion, not presented separately, as should the section titled “Interpretation and implications of the study”, which duplicates the discussion. I would suggest that the authors use subheadings more strategically throughout the Discussion section, to highlight the key messages and implications, and to reflect on opportunities to potentially strengthen the consultation approach so that public health considerations can be front and centre. - The sub-headings are as suggested in BMJ Open instructions to authors, which we have followed. **** end previous comment and response **** 3. No response at all was given to the following comment (other than a small amendment regarding the example). “Structuring the findings to address only the consultation questions separately I think minimises the analytical value of the study. This doesn’t seem to stray far from what an interested observer could see from the submissions and Ofcom’s response. I think the current examination of the policy change in relation to what stakeholders were requesting is essential, but that the study would add more value with additional inclusion of a cross-question analysis, that examines the core study question of how stakeholders influence policy.” This remains a weakness in the current presentation of Results.
--	---

VERSION 2 – AUTHOR RESPONSE

Reviewer: 1

Authors have done a great job responding to the critiques of the first round. Could they consider one more thing? The title is rather generic and could be made more interesting and specific to the issue and methods

Thank you for your comments.

We have changed the initial part of the title to ‘what arguments and from whom are most influential in shaping public health policy’ to better reflect our stated aims and be more specific to the issue as suggested by the reviewer.

Reviewer: 2

The paper has been strengthened in several regards, however, I still have some concerns. In particular, I do not see that the authors have engaged with the substance of the concerns that I raised in my previous review. Although the authors have acknowledged the major concern I raised regarding the conclusion, and amended the body of the article to make the public health influence clearer, the overall conclusion still states "Public health policy making may prioritise commercial over public health interests" rather than reflecting the findings of this study; the data as far as I understand it shows the influence of both public health and industry, and minimal 'watering down' of the policy. As per Table 1, it appears the main change from Package 1 (the strongest regulation proposed) to Amended Package 1 (which was adopted), was in fact to strengthen it in line with public health.

We have updated the initial line of the conclusion to read: ‘Public health policy-making appears to be considered as a balance between commercial and public health interests’ which reflects the conclusion we have reached, and the suggestions made by the reviewer.

The reviewer correctly states that the package of measures adopted shows influence from both public health and industry. The point we are making with the sentence 'commercial considerations appear to have led to a watering down of initial regulatory proposals' is slightly different – that measures that were advocated for by the public health community, such as the ban on television HFSS food advertising pre-9pm, were not even considered as options by Ofcom due to their perceived impact on revenue. Taking this into account, it does appear that commercial interests were able to frame the consultation as a balance between revenue and public health prior to it even beginning, and thus had a substantial impact on the outcome.

The conclusion goes on to state that 'commercial considerations appear to have led to a watering down of initial regulatory proposals', not the watering down of the eventual policy. The reviewer is correct in saying that Table 1 shows both influence from both public health and industry. However, this comment refers to the initial regulatory proposals, which were criticised in a number of responses for taking more robust public health measures off the table before the consultation had even started. We have added a further line to the abstract to further highlight this subtle but important difference.

Other unaddressed concerns are detailed below.

1. My query pasted below did not seem to be clear - what I was referring to was the need to note the differences between stakeholder groups in the table, not to add numbers (ie. "some respondents" doesn't give an indication of stakeholder group differences). This is in line with the articulated aim of the study (p6): "We aimed to identify which arguments, and from which stakeholder groups, appeared to be most influential in shaping the changes of Ofcom's position from the initial consultation to the final recommendations." The data as presented in Table 3 does not give any indication of which stakeholder groups made which arguments.

**** previous comment and response ****.

- Table on page 10 is not specific enough to be useful – the paper argues that there are differences by stakeholder groups and this should be clear in the summary table: e.g. "Some responses highlighted that children may watch adult TV... Other responses worried that this would disproportionately impact advertising revenues" (emphasis added)

- This was a qualitative analysis and, as such (and as stated in the aims), we were interested in the range of arguments made by different stakeholder groups and which of these appeared to be most influential in shaping eventual policy. We were not interested in the frequency with which different arguments were made and it was not our intention to conduct a quantitative analysis of the relationship between, for example, frequency of argument and likelihood of that argument shaping policy. Thus, we have chosen not to include specific numbers in the text, as we think this would detract from the qualitative nature of the enquiry and not add useful information.

**** end previous comment and response ****

Thank you for this clarification, that's a very good point. We have adjusted the table to reflect which stakeholder groups we are referring to.

2. There was no response given to the substantive suggestions in the following queries - in particular, the need for additional discussion that relates to international experience and a clearer logic throughout the discussion.

**** previous comment and response ****

10. I don't understand the relevance of the comment on self regulation from line 416 onwards, as this is not self regulation. Please provide a more nuanced discussion of this regulation in light of other international approaches instead.

- Comment on self-regulation removed.

**** end previous comment and response ****

We considered discussing self-regulation as suggested by the reviewer. After consideration, we felt that self-regulation was tangential to the purpose of the paper, which is focused on stakeholder influence on policy. Self-regulation is clearly an important topic. However, other papers explore this and we would not be able to provide a detailed and nuanced discussion of self-regulation in light of other international approaches within the context of this paper. As a result, we removed the comment.

**** previous comment and response ****

11. The section titled "Relationship to existing knowledge" from line 452 onwards should be integrated into the discussion, not presented separately, as should the section titled "Interpretation and implications of the study", which duplicates the discussion. I would suggest that the authors use subheadings more strategically throughout the Discussion section, to highlight the key messages and implications, and to reflect on opportunities to potentially strengthen the consultation approach so that public health considerations can be front and centre.

- The sub-headings are as suggested in BMJ Open instructions to authors, which we have followed.

**** end previous comment and response ****

Whilst we appreciate the reviewer's suggestion for a more integrated discussion and their personal preference for this approach, we have structured the discussion according to BMJ Open house style and this led to some duplication. We have attempted to reduce this duplication in this draft, for instance by removing lines 421-422 and 513-516. We will defer to the editor with regards to whether they would prefer us to stick to the house style or provide a more integrated discussion as requested by the reviewer.

3. No response at all was given to the following comment (other than a small amendment regarding the example).

"Structuring the findings to address only the consultation questions separately I think minimises the analytical value of the study. This doesn't seem to stray far from what an interested observer could see from the submissions and Ofcom's response. I think the current examination of the policy change in relation to what stakeholders were requesting is essential, but that the study would add more value with additional inclusion of a cross-question analysis, that examines the core study question of how stakeholders influence policy."

This remains a weakness in the current presentation of Results.

Reviewer 1 rightly pointed out that the original title was too broad as in this study we are not able to identify exactly how stakeholders influence policy as we have only explored one means of influencing policy. Therefore, we have adjusted the title of the paper to be more in-line with our aims, which were to explore which arguments, and from which stakeholder groups, appear to be influential in the eventual policy measures that Ofcom took. We have also added to the limitations section to highlight this:

'In this study, we have only addressed what arguments and from whom are most influential in shaping public health policy, not specifically the various methods by which different stakeholders influence policy.... Further work could explore other means of influence in due course. (Lines 478-480 and lines 485-486)

We hope that these changes make the core study question clearer. As the reviewer says, an additional approach would be to do a cross-question analysis. As mentioned in lines 199-202, most responses were broad statements rather than specific answers to the consultation questions. It was felt that a focus on the policy measures taken and how stakeholder groups influenced the shape of those would be a more effective approach in answering the core study question. We have added in the following statement to the limitations section (lines 476-477) to reflect that a cross-question analysis could be an option for further study:

'A cross-question analysis could have been performed analysing responses by each question posed, although many of the responses were free text and did not address each question directly'